# *OSCA* Genes in Bread Wheat: Molecular Characterization, Expression Profiling, and Interaction Analyses Indicated Their Diverse Roles during Development and Stress Response

**DOI:** 10.3390/ijms232314867

**Published:** 2022-11-28

**Authors:** Amandeep Kaur, Alok Sharma, Sameer Dixit, Kashmir Singh, Santosh Kumar Upadhyay

**Affiliations:** 1Department of Botany, Panjab University, Chandigarh 160014, India; 2Department of Biology, The University of Western Ontario, London, ON N6A 3K7, Canada; 3Department of Biotechnology, Panjab University, Chandigarh 160014, India

**Keywords:** abiotic, biotic, expression, miRNA, segmental, stress

## Abstract

The hyperosmolality-gated calcium-permeable channels (OSCA) are pore-forming transmembrane proteins that function as osmosensors during various plant developmental processes and stress responses. In our analysis, through in silico approaches, a total of 42 *OSCA* genes are identified in the *Triticum aestivum* genome. A phylogenetic analysis reveals the close clustering of the OSCA proteins of *Arabidopsis thaliana*, *Oryza sativa*, and *T. aestivum* in all the clades, suggesting their origin before the divergence of dicots and monocots. Furthermore, evolutionary analyses suggest the role of segmental and tandem duplication events (Des) and purifying selection pressure in the expansion of the *OSCA* gene family in *T. aestivum*. Expression profiling in various tissue developmental stages and under abiotic and biotic stress treatments reveals the probable functioning of *OSCA* genes in plant development and the stress response in *T. aestivum*. In addition, protein–protein and protein–chemical interactions reveal that OSCA proteins might play a putative role in Ca^2+^-mediated developmental processes and adaptive responses. The miRNA interaction analysis strengthens the evidence for their functioning in various biological processes and stress-induced signaling cascades. The current study could provide a foundation for the functional characterization of *TaOSCA* genes in future studies.

## 1. Introduction

Plants on Earth face several adverse environmental conditions throughout their lives and respond to them by adopting various strategies, such as the sensing and decoding of signals, which in turn initiate the downstream signaling cascades and expression of several stress-responsive genes [1,2]. Calcium signaling is one of the important phenomena that occur in plants and coordinates numerous developmental processes and stress responses. Cytosolic calcium ion (Ca^2+^) concentration levels usually remain in the range of 100–200 nanomolar (nM) during normal conditions [3]. However, they undergo a signal-specific transient rise during antagonistic environmental circumstances, including osmotic stress, which is known as the “Ca^2+^ signature” [2]. This Ca^2+^ signature is usually decoded by specific Ca^2+^ binding or sensor proteins that ultimately initiate the stress-related signaling pathways [4]. The cytosolic Ca^2+^ concentration is maintained by the simultaneous functions of three types of Ca^2+^ transporters, known as channels, exchangers, and pumps [5,6].

The hyperosmolality-gated calcium-permeable channels (OSCAs) are the membrane-localized Ca^2+^-permeable channels involved in the influx of Ca^2+^ from the apoplast to the cytoplasm. They are usually activated in response to hyper-osmotic conditions, dehydration, and mechanical stress in plants [7,8,9]. Similar Ca^2+^-permeable channels work as osmosensors in animals and bacteria [10,11]. The structural configuration has exhibited the occurrence of a conserved functional domain, i.e., DUF221 (Pfam accession: 02714) in OSCA proteins, which plays an important role in osmotic adjustment [12,13].

*OSCA* genes have been identified in various plant species, including *Arabidopsis thaliana* (15 genes); *Oryza sativa* L. *Japonica*, *Oryza sativa* L. ssp. *Indica*, and *Oryza brachyantha* (11 genes); *Oryza glaberrima* (12 genes); *Pyrus bretschneideri* (16 genes); *Vigna radiata* (13 genes); *Zea mays* (12 genes); *Gossypium hirsutum* (35 genes); *Gossypium arboretum* (21 genes); *Gossypium raimondii* (22 genes); and *Glycine max* (21 genes) [14,15,16,17,18,19,20]. Several studies have suggested the role of *OSCA* genes in various plant developmental processes and stress responses. For instance, Yaun et al. characterized OSCA1 in *A. thaliana* and revealed that it mediates an osmotic-stress-evoked Ca^2+^ concentration increase and plays a crucial role in osmosensing [14]. ZmOSCA2.4 in *Z. mays* provided drought stress tolerance in *A. thaliana* [21]. Another report suggested that the *OsOSCA* genes of *O. sativa* performed redundant functioning in Ca^2+^ signaling induced in response to hyperosmolality and salt stresses [22,23]. The GhOSCA1.1 of *G. hirsutum* enhances salt and drought resistance by modulating the activity of antioxidant enzymes [19]. Further, the role of OSCA1.3 and OSCA1.7 in Arabidopsis has been established in pathogen-associated molecular patterns (PAMPs)-triggered stomatal immunity [24,25]. These studies revealed the importance of OSCA channels in plants.

Due to the immense importance of *OSCA* genes in plants, there was a grave need for an inclusive characterization of the genes in an important staple food crop, *T. aestivum* L. Herein, a total of 42 *TaOSCA* genes were identified following various in silico approaches in the *T. aestivum* genome, which were used for further analyses, including duplication event, Ka/Ks, phylogenetic, gene, and protein structure analysis, using various bioinformatic approaches. To gain insight into the roles of *TaOSCA* genes, an expression analysis was carried out in various tissue developmental stages and biotic and abiotic stress conditions using high-throughput RNA seq data and qRT-PCR. Furthermore, protein–protein, protein–chemical interaction, and miRNA interaction analyses were performed to explore their diverse roles. This work could provide new directions for the functional characterization of TaOSCA proteins in future studies.

## 2. Results

### 2.1. In Silico Identification, Chromosomal Distribution, and Protein Characterization

A total of 42 *TaOSCA* genes were identified in the *T. aestivum* genome by performing a BLAST search using the already reported 15 and 11 *OSCA* sequences of *A. thaliana* and *O. sativa*, respectively. The identified *TaOSCA* genes could assemble into 14 homoeologous groups based on sequence similarity (≥90%) among them and their subgenomic (A, B and D) locations. Then, their nomenclature was conducted following the standard wheat symbolization protocol (Appendix A). The 11 homoeologous groups, including OSCA1, OSCA3-6, and OSCA9-14, consisted of genes from each A, B, and D subgenome. However, three homoeologous groups carried genes from only two subgenomes. For instance, the homoeologous group OSCA2 consisted of genes from the A and D subgenomes, while the OSCA7 and OSCA8 homoeologous groups consisted of genes from the B and D subgenomes. The highest number of *TaOSCA* genes were derived from subgenome D (15), followed by subgenome B (14) and subgenome A (13). A chromosomal distribution analysis showed that all the identified *TaOSCA* genes were scattered on 15/21 chromosomes, including 1A, 1B, 1D, 2A, 2B, 2D, 3A, 3B, 3D, 4A, 4B, 4D, 5A, 5B, and 5D. A maximum number of four genes were found on chromosomes 1A, 1D, 4B, and 5A each, while a minimum of one gene was found on chromosome 3A (Appendix A; Figure 1).

The in silico protein characterizations revealed the various important features of TaOSCA proteins. The peptide length of TaOSCA proteins varied from 469 to 804 amino acid (aa) residues (Appendix A). The longest protein was TaOSCA12-A, while the shortest was TaOSCA10-B2. Moreover, the molecular weight and isoelectric point of TaOSCA proteins ranged from 53.7 to 93.4 kDa and 6.59 to 9.77, respectively (Appendix A). Among all the TaOSCA proteins, the heaviest and lightest proteins were TaOSCA3-A and TaOSCA10-B2, respectively. Out of 42 OSCA proteins, the pI of 40 proteins was more than 7, while two proteins, TaOSCA6-B and TaOSCA6-D, had less than 7. The subcellular localization for all the TaOSCA proteins was predicted on the plasma membrane. None of the TaOSCA proteins consisted of any kind of signal peptide. However, the number of transmembrane helices ranged from five to eleven (Appendix A).

### 2.2. Prediction of Duplication Events and Ka/Ks Analysis

The paralogous genes originated at the time of duplication events led to the expansion of gene families. To understand the expanded progression of the *TaOSCA* gene family, the duplication events were analyzed. A total of three duplication events were predicted, which were segregated into two segmental duplication events (SDEs) (*TaOSCA4-A-TaOSCA6-B* and *TaOSCA1-B-TaOSCA2-D*) and one tandem duplication event (TDE) (*TaOSCA12-B-TaOSCA13-B*) in *T. aestivum* (Appendix A; Figure 1).

The Ka/Ks ratio was calculated to examine the type of selection pressure which affected the duplicated gene pairs during the evolution. A Ka/Ks ratio of less than one value suggests a negative selection (purifying), while a value of more than one shows the presence of positive selection pressure. In our analysis, the Ka/Ks ratio of all the identified duplicated gene pairs was less than one, which suggested that they had endured negative or purifying selection pressure. In addition to this, the time of divergence was also estimated, which was in a range from 13.0 to 32.9 MYA (Table 1).

By performing Tajima’s relative rate test, it was analyzed that all the duplicated gene pairs had a non-significant χ2-value at *p* > 0.05 (Table 2). According to Tajima, *p*-values of more than 0.05 point towards the acceptance of the molecular clock hypothesis, which signifies the equal rate of evolution among all the duplicated gene pairs [26].

### 2.3. Phylogenetic Analysis

To gain insight into the evolutionary connections of the OSCA proteins among *A. thaliana*, *O. sativa*, and *T. aestivum*, the phylogenetic tree was generated using full-length OSCA protein sequences. In the phylogenetic tree, these OSCA proteins were tightly clustered into four clades named I, II, III, and IV. Clade I and clade II consisted of 28 OSCA proteins, while clade III and clade IV consisted of 7 and 5 OSCA proteins, respectively (Figure 2). A total of eight AtOSCAs, four OsOSCAs, and sixteen TaOSCAs were clustered in clade I, while clade II consisted of five OSCA proteins each from *A. thaliana* and *O. sativa* and eighteen OSCA proteins from *T. aestivum*. One AtOSCA and one OsOSCA were present in each of the clades III and IV, while a total of five and three TaOSCA proteins were clustered in clades III and IV, respectively (Figure 2).

All the homologous genes of *T. aestivum*, for example, *TaOSCA13-A*, *TaOSCA13-B*, *and TaOSCA13-D; TaOSCA1-A*, *TaOSCA1-B*, and *TaOSCA1-D*, etc., were closely clustered to each other in the phylogenetic tree. In addition to this, all the identified paralogous genes of *T. aestivum* were also clustered nearby, for instance, *TaOSCA4-A-TaOSCA6B*, *TaOSCA12-B-TaOSCA13*, and *TaOSCA1-B-TaOSCA2-D*, etc. (Figure 2).

### 2.4. Expression Profiling of TaOSCA Genes in Tissue Developmental Stages

We analyzed the expression patterns of *TaOSCA* genes in three developmental stages of five tissues, including root_z10, root_z30, root_z39, stem_z30, stem_z32, stem_z65, leaf_z10, leaf_z23, leaf_z71, spike_z32, spike_z39, spike_z65, grain_z71, grain_z75, and grain_z85, using available high-throughput RNA-seq data [27,28]. Only those *TaOSCA* genes were chosen for further analysis which exhibited ≥2 FPKM values in at least one tissue development stage. Some of the *TaOSCA* genes showed specifically higher expression in the specific developmental stages of root, shoot, leaf, spike, and grain tissues (Figure 3). For instance, *TaOSCA1-A*, *TaOSCA1-D*, *TaOSCA7-B*, and *TaOSCA7-D* exhibited significant expression levels in almost all the tissue developmental stages. *TaOSCA1-A* and *TaOSCA1-D* were found to be highly expressed in grain_z71; leaf_z71; root_z10; leaf_z23; spike_z32 and spike_z39; and stem_z32. *TaOSCA7-B* and *TaOSCA7-D* showed higher expression levels in the early stages of roots, stems, leaves, spikes, and all the stages of grains. Moreover, some genes, such as *TaOSCA4-D*, *TaOSCA4-A*, *TaOSCA4-B*, and *TaOSCA8*-B, etc., exhibited no or less expression in tissue developmental stages (Figure 3).

### 2.5. Expression Profiling of TaOSCA Genes in Abiotic Stresses

To understand the putative role of *TaOSCA* genes in abiotic stress responses, their expression patterns were analyzed under heat (HS), drought (DS), combined heat–drought (HD), and salt stress using available high-throughput RNA sequence data [29,30]. Those *TaOSCA* genes were selected for analysis which showed more than two-fold down- or upregulation in at least one of the abiotic stress conditions. This criterion led to the identification of a total of 31 *TaOSCA* genes, which displayed differential expression in response to the DS, HS, and HD stress conditions (Figure 4A). The majority of *TaOSCA* genes were found to be upregulated after DS treatments, especially after DS 1 h. However, some genes, such as *TaOSCA3-D*, *TaOSCA11-B*, *TaOSCA11-D*, *TaOSCA10-B1*, etc., showed upregulation at the later stage (6 h) of HS and HD. However, only a few *TaOSCA* genes, such as *TaOSCA6-D* and *TaOSCA12-B*, etc., exhibited upregulation at an early hour (1 h) of the HS and HD conditions. *TaOSCA12-B* displayed significant upregulation after every hour of the DS, HS, and HD treatments (Figure 4A) and showed maximum expression (6-fold) after 6 h of DS. Furthermore, *TaOSCA8-B* was highly upregulated (69-fold) at DS 6 h as compared to the rest of the genes. The genes *TaOSCA3-A*, *TaOSCA3-B*, *TaOSCA7-B*, *TaOSCA10-B2*, *TaOSCA10-D*, and *TaOSCA13-A* were downregulated after each hour of the DS, HS, and HD treatments (Figure 4A).

Out of 42 genes, 29 *TaOSCA* genes showed differential expression in response to salt stress. The majority of the *TaOSCA* genes were found to be upregulated in the early hours of salt stress (Figure 4B). However, some genes, such as *TaOSCA11-B*, *TaOSCA12-A*, *TaOSCA12-D*, *TaOSCA14-A*, *TaOSCA14-B*, and *TaOSCA14-D*, were upregulated after 6, 12, 24, and 48 h of salt stress. Moreover, it was predicted that at 6, 24, and 48 h of salt stress, *TaOSCA5-A2* would be highly upregulated (11, 18, and 9-fold), and at 12 h of salt stress, *TaOSCA14-B* would be highly upregulated (10-fold) as compared to other genes (Figure 4B). Some genes, such as *TaOSCA2-D*, *TaOSCA3-A*, *TaOSCA3-B*, *TaOSCA6-B*, *TaOSCA7-B*, *TaOSCA7-D*, *TaOSCA10-A*, *TaOSCA13-A*, and *TaOSCA13-B*, were down-regulated at each hour of salt stress.

### 2.6. Expression Profiling of TaOSCA Genes in Biotic Stresses

To determine whether the expression of *TaOSCA* genes was responsive to biotic stress, their expression was analyzed against two fungal pathogens, i.e., *Blumeria graminis* (bgt) and *Puccinia striiformis* (Pst), using high-throughput RNA seq data [31]. Out of 42, a total of 28 *OSCA* genes were differentially expressed after Bgt and Pst infestation (Figure 4C). The majority of *OSCA* genes were significantly upregulated after Bgt infestation, for instance, *TaOSCA14-B*, *TaOSCA14-D*, *TaOSCA11-D*, *TaOSCA10-A*, and *TaOSCA10-B2*, etc. However, a few genes, such as *TaOSCA2-D*, *TaOSCA3-B*, *TaOSCA11-B*, *TaOSCA7-B*, and *TaOSCA7-D*, etc., also exhibited upregulation at different hours of Pst infestation. The gene *TaOSCA11-A* was found to be differentially expressed at each hour of Bgt and Pst inoculation (Figure 4C). Furthermore, it was analyzed that some genes, such as *TaOSCA1-A*, *TaOSCA1-B*, *TaOSCA1-D*, *TaOSCA5-A1*, *TaOSCA10-B1*, *TaOSCA10-D*, *TaOSCA12-D*, *TaOSAC13-B*, and *TaOSCA13-D*, were downregulated after each hour of Bgt and Pst infestation.

### 2.7. qRT-PCR Analysis under CaCl_2_ Treatment

Since the OSCAs are Ca^2+^-permeable channels and are involved in the import of Ca^2+^ in the cytoplasm of plant cells that ultimately leads to Ca^2+^ signaling [24,25], an expression analysis of six *TaOSCA* genes representing each phylogenetic clade (i.e., *TaOSCA1-D*, *TaOSCA3-B*, *TaOSCA5-A1*, *TaOSCA11-D*, *TaOSCA12-D*, and *TaOSCA14-A*) was performed after CaCl_2_ treatment using qRT-PCR (Appendix A). The gene *TaOSCA1-D* showed upregulated expression at 12 h of CaCl_2_ treatment, and afterwards expression decreased (Figure 5). The gene *TaOSCA3-B* exhibited negligible expression at each hour of CaCl_2_ treatment. However, *TaOSCA5-A1* and *TaOSCA12-D* were found to be upregulated at 24 h of CaCl_2_ treatment, while two genes, *TaOSCA11-D* and *TaOSCA14-A*, showed significant expression at the initial hour of CaCl_2_ treatment (Figure 5).

### 2.8. Co-Expression Analysis and Gene Ontology Mapping

To reveal the putative mechanisms of action of *TaOSCAs*, we performed the co-expression of *TaOSCAs* with other genes of *T. aestivum* using the expression data of tissue developmental stages and under biotic and abiotic stresses. During tissue development, a total of ten *TaOSCA* genes were found to be co-expressed with 49 transcripts (Appendix A; Figure 6A). These co-expressed transcripts encoded serine/threonine-protein kinase, peroxidase 4-like, embryogenesis-associated protein, cyclin-dependent kinase, UDP-glucose 6-dehydrogenase 4-like, GATA transcription factor 20-like, and meiotic recombination protein, etc. The gene ontology (GO) mapping of the identified co-expressed transcripts revealed their putative roles in several biological processes, such as the oxidative photosynthetic carbon pathway (GO:0009854), glycosaminoglycan biosynthetic process (GO:0006024), diterpenoid biosynthetic process (GO:0016102), cellular lipid metabolic process (GO:0044255), pollen development (GO:0009555), seed development (GO:0048316), integral component of membrane (GO:0016021), and UDP-glucuronate biosynthetic process (GO:0006065), etc.

During drought and heat stresses, a total of 11 *TaOSCA* genes were co-expressed with 120 transcripts of *T. aestivum* (Appendix A; Figure 6B). These transcripts encoded rho GTPase-activating protein 7-like isoform X3, calcium-dependent protein kinase 17, universal stress protein YxiE-like, probable aquaporin TIP3-2, auxin response factor 4, WRKY transcription factor 3, calcium uptake protein, and ABC transporter E family member 2, etc. The GO mapping of these transcripts showed that they are mainly involved in defense response (GO:0006952), response to stimulus (GO:0050896), protein kinase activity (GO:0004672), water transport (GO:0006833), water channel activity (GO:0015250), oxidoreductase activity (GO:0016491), signal transduction (GO:0007165), peptidyl-serine phosphorylation (GO:0018105), calcium ion binding (GO:0005509), and dGTPase activity (GO:0006203), etc.

In the presence of salt stress, a total of 26 *TaOSCA* genes showed co-expression with 116 transcripts of *T. aestivum* (Appendix A; Figure 6C). These transcripts encoded ABC transporter G family member 16-like, calcium-transporting ATPase 5, serine/threonine protein phosphatase 2A, transcription factor SRM1-like, trans-cinnamate 4-monooxygenase, chaperone protein dnaJ 20, and ribulose-1,5 bisphosphate carboxylase/oxygenase large subunit N-methyltransferase, etc. The GO mapping suggested their role in the lignin metabolic process (GO:0009808), salicylate 1-monooxygenase activity (GO:0018658), L-glutamate transmembrane transport (GO:0015813), calcium ion transmembrane transport (GO:0070588), iron ion binding (GO:0005506), the response to photooxidative stress (GO:0080183), defense response (GO:0006952), ABC-type transporter activity (GO:0140359), electron transport chain (GO:0022900), and the cell surface receptor signaling pathway (GO:0007166), etc.

In the presence of biotic stress, a total of 11 *TaOSCA* genes were found to be co-expressed with the 84 transcripts of *T. aestivum* (Appendix A; Figure 6D), which encoded disease resistance RPP13-like protein 4, chitinase CLP-like, disease resistance protein PIK6-NP-like, pathogenesis-related thaumatin-like protein 3.5, wall-associated receptor kinase 2-like isoform X1, calcium-binding protein 39-like, myb-related protein MYBAS1-like isoform X2, and E3 ubiquitin-protein ligase AIRP2-like, etc. The GO mapping suggested their probable functioning as hydrolase activity (GO:0016787), ncRNA processing (GO:0034470), catalytic complex (GO:1902494), intracellular signal transduction (GO:0035556), protein phosphorylation (GO:0006468), polysaccharide binding (GO:0030247), defence response to fungus (GO:0008843), metal ion binding (GO:0046872), and ubiquitin-protein ligase activity (GO:0061630), etc.

### 2.9. Protein–Protein Interaction Analysis

The protein–protein interaction analysis using the STRING server suggested the interaction of TaOSCAs with 13 different proteins: AIR3 (subtilisin-like serine endopeptidase), ARFC1 (ADP-ribosylation factor C1), LEA4-1 (late embryogenesis abundant 4-1), C/VIF2 (cell wall/vacuolar inhibitor of fructosidase 2), CYP71B5 (cytochrome p450 71b5), FLA6 (fasciclin-like arabinogalactan protein 6), GLR2.3 (glutamate receptor 2.3), GRV2 (gravitropism defective 2), NiaP (organic cation/carnitine transporter 7), PLP4 (patatin-like protein 4), UPS4 (ureide permease 4), VSR1 (vacuolar-sorting receptor 1), and YLMG1-2 (YGGT family protein) (Appendix A; Figure 7A). Moreover, it was found that a total of 29, 28, and 26 TaOSCAs interacted with C/VIF2, AIR3, and UPS4 protein, respectively. Six TaOSCA proteins, TaOSCA4-A, TaOSCA4-B, TaOSCA4-D, TaOSCA6-A, TaOSCA6-B, and TaOSCA6-D, showed interactions with CYP71B5, FLA6, GLR2.3, NiaP, and PLP4. The five OSCA proteins TaOSCA5-A1, TaOSCA5-B, TaOSCA5-D1, TaOSCA5-A2, and TaOSCA5-D2 were found to have interacted with LEA4-1 and YLMG1-2, while three TaOSCA proteins, TaOSCA11-A, TaOSCA11-B, and TaOSCA11-D, showed interaction with ARFC1, GRV2, and VSR1 proteins (Appendix A; Figure 7A).

### 2.10. Protein–Chemical Interaction Analysis

A protein–chemical interaction analysis suggested the interaction of 37 TaOSCA proteins with 15 different chemicals: 5-hydroxytryptophan, beta-D-glucopyranoside, AR-11, AR-21, cytidine triphosphate, emodin anthrone, glucose, guanosine triphosphate, hyperforin, isobutyrophenone, maltose solution, MgATP, pyrophosphate, Tyr-Met-Lys, and uridine 5′-triphosphate (Appendix A; Figure 7B). Furthermore, it was analysed that 31 TaOSCA proteins interacted with beta-D-glucopyranoside, cytidine triphosphate, glucose, guanosine triphosphate, maltose solution, MgATP, pyrophosphate, and uridine 5′-triphosphate. The three proteins TaOSCA14-A, TaOSCA14-B, and TaOSCA14-D showed interaction with 5-hydroxytryptophan, AR-11, AR-21, emodin anthrone, hyperforin, and isobutyrophenone, while the three proteins TaOSCA3-A, TaOSCA3-B, and TaOSCA3-D were found to have interacted with Tyr-Met-Lys (Appendix A; Figure 7B).

### 2.11. miRNAs Interaction Analysis

A total of 27 miRNAs were found to interact with 28 *OSCA* transcripts of *T. aestivum* (Appendix A; Figure 8A,B). Out of these 27 identified miRNAs, 22 miRNAs acted via cleavage and 4 miRNAs acted via translation inhibition with their targeted transcripts. Moreover, it was analyzed that only one miRNA, i.e., ta-miR2038a, acted via both translation inhibition and the cleavage mode of their target transcripts. Furthermore, it was predicted that a single *TaOSCA* transcript would be targeted by multiple miRNAs, for instance, *TaOSCA7-B* by ta-miR2004a, ta-miR2012a, ta-miR129a, ta-miR2038a, and tae-miR1132a, and *TaOSCA12-D* by tae-miR156h, tae-miR528a, tae-miR530a, and ta-miR090a. In addition to this, we observed that multiple *TaOSCA* transcripts were targeted by the same miRNA; for instance, *TaOSCA12-A*, *TaOSCA12-B*, *TaOSCA12-D*, *TaOSCA13-A*, *TaOSCA13-B*, *TaOSCA13-D*, *TaOSCA5-A1*, *TaOSCA5-B*, and *TaOSCA5-D1* were targeted by tae-miR530a, and *TaOSCA1-A*, *TaOSCA1-B*, *TaOSCA1-D*, *TaOSCA2-D*, *TaOSCA7-B*, and *TaOSCA7-D* were targeted by ta-miR2038a.

## 3. Discussion

OSCA proteins are very important Ca^2+^-permeable channels reported to be involved in various stress responses as well as in the development process of plants. They have been identified and characterized in numerous plant species at different scales [12,13,14,15,16,17,18,19,20,21,22,23,24,25]. However, the detailed characterization of TaOSCAs in *T. aestivum*, an important cereal crop, has not been performed in earlier studies. Moreover, while Tong et al. [32] earlier identified a similar number of *TaOSCA* genes as were identified in the present study, they only conducted a partial characterization and expression analysis of a limited number of genes in different experimental conditions. In the current study, a detailed in silico characterization, along with an evolutionary analysis, expression profiling in diverse conditions, and interaction analyses were performed.

The extensive BLAST search identified a total of 42 *TaOSCA* genes in the allohexaploid genome of *T. aestivum*, which were relatively higher and around thrice in number than in other diploid plant species, such *as A. thaliana*, *O. sativa*, and *Z. mays*, etc. Moreover, a relatively higher number of *OSCA* genes has also been reported in tetraploid *G. hirsutum* (35) [14,15,16,17,18,19,20]. These results suggested that the occurrence of a higher number of *OSCA* genes could be attributed to their complex polyploid genome. Further, a positive correlation between the number of genes and the ploidy level of the genome was observed. The comparable number of the homeologous grouping (14) of the *TaOSCA* genes of *T. aestivum* to the total number of *OSCAs* in the diploid plant species [14,15,17,18] further endorses the above hypothesis. Similarly, higher numbers of genes in other gene families of *T. aestivum* have also been reported as compared to those of other crops [33,34]. The other characteristics of the TaOSCA proteins, such as peptide length, molecular weight, pI, sub-cellular localization, and the number of trans-membranes, were comparable to the OSCA proteins of other plant species [14,15,16,17,18,19,20]. The majority of TaOSCA proteins were predicted to be localized in the plasma membrane, which was also confirmed by the presence of the trans-membrane domain in each protein.

Duplication events play vital functions in the expansion of gene families, introducing genetic variability, and gaining neo-functionality in paralogues genes [35]. In *T. aestivum*, both SDEs and TDEs were predicted as key factors behind the expansion of the *TaOSCA* gene family. Similar duplication events in the *OSCA* gene family have also been reported in *Pyrus bretschneideri* (five SDEs), *V. radiata* (three SDEs), and *G. hirsutum* (16 SDEs) [16,17,19]. During evolution, the paralogous genes experience positive (non-purifying) or negative (purifying) selection pressures under various evolutionary forces [36]. The Ka/Ks analysis suggested the purifying selection of the *TaOSCA* paralogous genes, which was in agreement with the paralogous *OSCA* genes in other plants, such as *Pyrus bretschneideri*, *V. radiata*, and *G. hirsutum*, [16,17,19]. Further, the divergence time analysis suggested the origin of duplicated *TaOSCA* genes probably earlier than the hybridization events of the A, B, and D subgenomes [37].

The phylogenetic analysis suggested the clustering of OSCA proteins into four clades as reported in earlier studies [18,19,32]. The tight clustering of the *OSCA* genes of *A. thaliana*, *O. sativa*, and *T. aestivum* in all the clades indicated their evolvement before the division of dicotyledons and monocotyledons. The tight clustering of the homeologous genes of *T. aestivum* could be due to their high sequence homology. Moreover, the close clustering of paralogous *TaOSCA* genes pointed toward their conserved nature, which was in accordance with the purifying selection.

Expression studies on the *TaOSCA* genes were performed to reveal their roles in the growth and development of vegetative and reproductive tissues, as well as in the stress response. The specifically higher expression levels of certain genes in particular tissue developmental stages suggested their precise roles in said tissue development. For instance, the higher expression of the majority of *TaOSCA* genes in the early developmental stages of root tissue indicated their vital function in Ca^2+^ absorption, which is highly required during the early stage for the proper growth and development of plants because Ca^2+^ is an integral part of the cell wall and membrane and also plays a significant role in various organelles and in signaling [6]. Similarly, the higher expression levels of a few *TaOSCA* genes in leaf and stem tissues further indicated their roles in vegetative tissue development. Moreover, the specifically greater expression levels of certain *TaOSCA* genes in particular developmental stage spike and grain tissue indicated their involvement in reproductive tissue development and the reproductive health of plants. Similarly, the differential expression of *OSCA* genes has also been reported in the cereal crop rice, where certain genes, such as *OsOSCA1.1*, *OsOSCA1.2*, *OsOSCA2.4*, *OsOSCA3.1*, and *OsOSCA4.1*, exhibited higher expression levels in the roots, shoots, mature stems, mature flag leaves, stamens, pistils, and mature seeds [15]. The specifically high expression levels of some of the *TaOSCA* genes in various developmental stages of grain indicated their roles during grain filling. This could be due to either the positive involvement of Ca^2+^ during grain development or the fact that increased osmotic potential is responsible for the enrichment of *TaOSCA* transcripts. Earlier research has also suggested the roles of several *OSCA* genes in *O. sativa* in caryopsis development and seed imbibition [15]. Further, the differential expression of *OSCA* genes in vegetative and reproductive tissues has also been reported in *P. bretschneideri* [16]. Moreover, the specific roles of each *OSCA* need to be established individually in future studies.

The modulated expression of various *TaOSCA* genes under abiotic stresses, such as heat, drought, salt, and biotic stresses including fungal (Bgt and Pst) infestation, suggested the putative roles of *TaOSCA* genes in defense responses. The majority of *TaOSCA* genes exhibited differential responses during drought and salinity stresses, which is in agreement with previous reports [16,17,18,19,20,32] and the mechanosensitive nature of these genes [7,13]. For instance, an earlier survey established that five *OSCA* genes in *O. sativa*, *OsOSCA1.2*, *OsOSCA2.1*, *OsOSCA2.4*, *OsOSCA2.5*, and *OsOSCA3.1*, showed upregulation after ABA, drought, PEG, and salinity treatments. Further, *OsOSCA4.1* is upregulated after PEG, salinity, and ABA treatments, while *OsOSCA1.1* is solely upregulated after PEG and salinity stress. The OSCA1 of *A. thaliana* has also been reported to play a precise role in osmotic stress-induced Ca^2+^ signaling [14]. It has also been established that the over-expression of various *OsOSCA* genes (*OsOSCA1.1-4*, and *OsOSCA2.1-2)* mediated Ca^2+^ signalling in *osca1* mutant of *A. thaliana* during hyperosmolality and salt stress [22,23]. Similarly, most of the *ZmOSCA* genes of *Z. mays* are upregulated in response to ABA, PEG, and NaCl treatments, and *ZmOSCA2.3* and *ZmOSCA2.4* exhibited significantly higher upregulation under these stresses [21]. Further, the over-expression of *ZmOSCA2.4* conferred drought resistance in the mutant of *A. thaliana* [21]. The elevated expression of certain *TaOSCA* genes in *T. aestivum* in response to biotic stresses has suggested their involvement in biotic stress-induced Ca^2+^ signaling. Thor et al. [24] recently established the BIK1-mediated activation of OSCA1.3 and OSCA1.7 after phosphorylation for Ca^2+^ influx in the cytoplasm of guard cells in Arabidopsis during PAMPs-triggered stomatal immunity, which established their role in Ca^2+^ signaling under biotic stress. The induced expression of *TaOSCA* genes at different hours of CaCl_2_ treatment in the present study further suggests their crucial participation in Ca^2+^ homeostasis and signaling. Several previous studies already established the roles of OSCAs in Ca^2+^ homeostasis in response to different stress conditions [14,15,16,17,18,19,20,21,22,23,24,25].

The interaction analyses of *TaOSCAs* were analysed by co-expression, protein–protein interaction, and protein–chemical interaction to reveal their roles in downstream signaling during developmental stages, metabolism, and stress conditions. The co-expression analysis and GO mapping indicated the involvement of *TaOSCA* genes in a wide range of developmental processes as well as in stress responses. For instance, the co-expression of *TaOSCA* genes with UDP-glucose 6-dehydrogenase 4-like, cellulose synthase, embryogenesis-associated proteins, and Rubisco revealed their role in cell wall synthesis, reproductive development, and photosynthesis [38,39]. During drought and heat stress, the co-expression of *TaOSCA* genes with Ca^2+^-binding protein, Ca^2+^ uptake proteins, Ca^2+^-dependent protein kinase, Ca^2+^ transporter ATPase, and WRKY transcription factor showed their role in Ca^2+^-related signaling cascades and Ca^2+^ homeostasis during these stresses [6,40]. The occurrence of *cis*-regulatory element DRE1 (drought-responsive element 1) in the promoter region of *TaOSCA* genes also indicated their functioning in drought and osmotic stress responses [32]. The co-expression of *TaOSCA* genes with transcripts encoding various kinases, such as serine-threonine kinase, well-associated receptor kinases, receptor-like kinases, and phosphatidylinositol 4-phosphate 5-kinase, further suggested their role in the defense-related signaling cascades [41]. During biotic stresses, the co-expression of *TaOSCA* genes with chitinase CLP-like and pathogenesis-related thaumatin-like protein 3.5 indicated their involvement in Ca^2+^-mediated downstream signaling for the activation of defense-responsive genes during biotic stresses [42], which was consistent with the modulated expression of certain *TaOSCA* genes after Bgt and Pst infestation.

Similar to the co-expression analysis, the protein–protein interaction analysis of TaOSCAs also suggested their linkage with various biological as well as stress-related signaling pathways. For instance, the interaction of TaOSCA proteins with GLR showed their functioning in stomata closure via Ca^2+^ influx into the cytosol [24,43], because both OSCA and glutamate receptors are known to be involved in Ca^2+^-mediated stomatal closure. The interaction of TaOSCA proteins with LEA and C/VIF2 proteins indicated their role in adaptive responses during drought stress conditions [44,45,46]. Additionally, their interaction with PLP4 and AIR3 suggested their role in root formation [47,48]. The significantly higher expression levels of various *TaOCSA* genes in root developmental stages also endorse the above statement. The interaction of TaOSCA with MgATP further suggested their probable functioning in Ca^2+^ signaling and homeostasis [49]. MgATP is reported to play an important role in plasma membrane ATPase and nitrogenase catalysis [50,51]. Interaction with GTPases that are responsible for oscillating Ca^2+^ concentration by modulating endo/exocytosis or by signaling cascades during pollen tube growth indicated their function in reproductive tissue development [52], which is also suggested by the specifically higher expression levels of certain *TaOSCA* genes in spike developmental stages. Guanosine triphosphate was also reported to be involved in signal transduction, vesicular transportation, and cell proliferation, etc. [52].

miRNAs control gene expression at the post-transcriptional and translational level through the regulatory mechanism known as RNA interference [53,54]. Many previous analyses have revealed the role of miRNAs in the regulation of several genes that are involved in plant developmental processes and stress responses [55,56,57,58]. Therefore, to analyze the miRNA-mediated regulation of *TaOSCA* genes, their interaction with the known miRNAs [59] of *T. aestivum* was investigated in the present study. The interaction of *TaOSCAs* with miR156 and miR172, associated with grain development [60] and floral organ development [61], respectively, further suggested the role of *TaOSCA* genes in plant reproductive development as indicated by expression and co-expression studies. Additionally, their interaction with some drought stress-related miRNAs, such as miR528, miRNA172, miRNA395, and miRNA530 [62,63,64], assured their miRNA-mediated regulation during the stress response.

In conclusion, comprehensive analyses of the *TaOSCA* gene family in *T. aestivum* suggested their diverse roles from development to the stress response, which could be attributed to their involvement in Ca^2+^ homeostasis. The duplication event analysis suggested that segmental duplications are the major factors responsible for the *OSCA* gene family expansion. The phylogenetic analysis of OSCA proteins among different plant species indicated the close evolutionary relatedness between dicotyledons and monocotyledons. The modulated expression patterns of *TaOSCA* genes suggested that they play essential roles in plant development as well as in adaptive responses. The co-expression and interaction analysis suggested the involvement of *TaOSCA* genes in Ca^2+^-mediated developmental processes and stress responses. *TaOSCA* interaction with numerous known miRNAs associated with different biological functions suggested the miRNA-mediated regulation of *TaOSCA* genes during different biological processes. The current study revealed numerous functions of TaOSCAs, but the specific roles of each of the genes need to be further established in future research.

## 4. Materials and Methods

### 4.1. Identification and Chromosomal Distribution

The bidirectional BLAST hit approach was adopted with e-value 10^−10^ to identify the *TaOSCA* genes in the *T. aestivum* genome. The OSCA protein sequences of *A. thaliana* and *O. sativa* were used as queries against the protein model sequences of *T. aestivum*, retrieved from the Ensembl Plant (http://plants.ensembl.org/index.html, accessed on 7 August 2021) and IWGSC (IWGSC RefSeq assembly v2.0) (http://wheat-urgi.versailles.inra.fr/Seq-Repository/Genes-annotations, accessed on 25 February 2019, http://www.wheatgenome.org/, accessed on 25 February 2019) databases. Furthermore, the SMART server [65] and Conserved Domain Database (CCD) blast search [66] approaches were used to confirm the presence of characteristic domains in the identified putative OSCA proteins of *T. aestivum*.

The data regarding the chromosomal and subgenomic location of each *TaOSCA* gene were downloaded from the Ensembl Plant server. Further, *TaOSCA* genes were mapped on their respective chromosomes using the MapInspect tool (http://mapinspect.software.in-former.com/accessed on 1 October 2021). The proposed guidelines for wheat gene symbolization (http://wheat.pw.usda.gov/ggpages/wgc/98/Intro.htm, accessed on 6 October 2021) were followed for the nomenclature of the TaOSCA proteins. The homeologous grouping of *TaOSCA* genes was performed on the basis of ≥90% sequence similarity among them, as performed in the previous analysis [33,34].

### 4.2. In Silico Protein Characterization

The Expasy tool was used to compute the various properties, for instance, the protein length, MW, and pI of each TaOSCA protein [67]. Subcellular localization, signal peptide, and transmembrane helices were predicted using the Cello [68], SignalP 5.0. (https://services.healthtech.dtu.dk/service.php?SignalP-5.0, accessed on 30 August 2022) and TMHMM-2.0 (https://services.healthtech.dtu.dk/service.php?TMHMM-2.0, accessed on 29 June 2022) server, respectively.

### 4.3. Prediction of Duplication Events and Ka/Ks Analysis

To analyze the duplication events, a sequence similarity matrix was conducted using the MAFFT software (https://www.ebi.ac.uk/Tools/msa/mafft/, accessed on 3 June 2022), and *TaOSCA* genes with 80–90% sequence similarity were considered duplicated genes. Further, the predicted duplicated genes were categorized into tandem and segmentally duplicated genes based on the distance among them as described in earlier reports [42].

The non-synonymous substitution per non-synonymous site (Ka), synonymous substitution per synonymous site (Ks), and Ka/Ks ratios of *TaOSCA* duplicated gene pairs were also calculated using the PAL2NAL software [69]. Further, the formula T = Ks/2r, (T = divergence time and r = divergence rate) was applied to compute the divergence time of *TaOSCA* duplicated genes. In the case of cereals, the value of r was proposed as 6.5 × 10^−9^ [70]. Moreover, Tajima’s relativity test was applied to estimate the evolutionary rate between duplicated genes [26].

### 4.4. Phylogenetic Analysis

For the construction of a phylogenetic tree, firstly full-length OSCA protein sequences of *A. thaliana*, *O. sativa*, and *T. aestivum* were aligned via the ClustalW algorithm (https://www.ebi.ac.uk/Tools/msa/clustalw2/, accessed on 9 June 2022). Further, the neighbor-joining method was applied to construct the phylogenetic tree OSCA protein using the MEGA X software with 1000 bootstrap replicates [71].

### 4.5. Expression Profiling of TaOSCA Genes

An expression analysis of the *TaOSCA* genes was performed to gain insight into the functional involvement of these genes in plant development and defense. The high-throughput RNA-seq data were downloaded from the publicly available URGI at https://urgi.versailles.inra.fr/files/RNASeqWheat, accessed on 24 February 2019. In the case of tissue developmental expression profiling, the RNA seq data were produced from three development stages of five tissues, including root, leaf, stem, spike, and grain tissues, and were present in biological replicates [27,28]. The trinity pipelines [72] were pursued to obtain fragments per kilobase of transcripts per million mapped (FPKM) reads, which were revalidated by the Expression ATLAS [73].

To study the effects of abiotic stresses on *TaOSCA* genes, expression profiling was carried out using high-throughput RNA-seq data generated after 1 and 6 h of HS, DS, and HD in duplicates from leaf tissue [29], and 6, 12, 24, and 48 h of salt stress (150mM NaCl) from root tissue [30].

Moreover, to study the effects of the two fungal pathogens Bgt and Pst on *TaOSCA* genes, expression profiling was carried out using high-throughput RNA seq data generated in triplicates after 24, 48, and 72 h of their inoculation [31]. Using the Trinity package, the expression values were calculated into FPKM reads [72]. The heat maps were built with the help of Hierarchical Clustering Explorer 3.5 (http://www.cs.umd.edu/hcil/hce/, accessed on 4 January 2022) and the Euclidean distance method was used for hierarchical clustering [74].

### 4.6. qRT-PCR Analysis

Firstly, the seeds of *T. aestivum* (cv. Chinese spring) were surface sterilized with sodium hypochlorite (1.2%) in 10% ethanol and washed with double autoclaved water. The washed seeds were placed overnight at 4 °C for stratification. Further, these seeds were kept at room temperature for germination. Afterwards, these germinated seedlings were grown in a plant growth chamber at 22 °C, 60% relative humidity, and in 16 and 8 hours of light and dark periods, respectively. Further, the seven-day-old seedlings were treated with CaCl_2_ (20 mM) Murashige and Skoog (MS) growth media. The samples were collected after 6, 12, 24, and 48 h in liquid nitrogen. RNA was taken out of root tissues using the Spectrum TM Plant Total RNA kit (Sigma, USA). The TURBO DNA-free™ Kit (Invitrogen, USA) was used to remove the traces of DNA contamination from the RNA samples. Moreover, both the quality and quantity of RNA samples were examined using agarose gel electrophoresis and a Nanodrop spectrophotometer, respectively. The cDNAs were produced from 1 microgram of RNA using Superscript III First-Strand Synthesis SuperMix (Invitrogen, Waltham, MA, USA). qRT-PCR was performed with CFX96 Real-Time PCR (BioRad, USA) using SYBR Green and gene-specific primers of selected *OSCA* genes by following a previously established procedure [75]. An ADP-ribosylation factor (TaARF1) was used as an internal control and the expression values were calculated using delta-delta CT (2-ΔΔCT) method [76]. All the experiments were carried out in biological triplicates (*n* = 3) which were further illustrated in the terms of mean § standard deviation (SD).

### 4.7. Co-Expression Analysis and Gene Ontology Mapping

The co-expressing partners of *TaOSCA* genes were predicted using CoExpress v.1.5. server. The Pearson correlation coefficient with a correlation power of 1 [77] and threshold filter ≥ 0.9 were used to compute the co-expression values. The Blast2GO tool [78] was used for the gene ontology (GO) mapping and functional annotation of co-expressed genes. Interaction networks of co-expressed genes were displayed using the Gephi 0.9.1 server [79].

### 4.8. Protein–Protein, Protein–Chemical, and miRNA Analysis

The protein–protein and protein–chemical interaction were performed using the STRING (http://stringdb.org) [80] and STITCH (http://stitch.embl.de/, accessed on 16 June 2022) [81] servers, respectively. Further, Cytoscape software (https://cytoscape.org/download.html) was used for the construction of interaction networks. The interacting miRNAs of *TaOSCA* genes were predicted with the psRNATarget (http://plantgrn.noble.org/psRNATarget/, accessed on 23 June 2022) database [82] using the known *T. aestivum* miRNA sequences [59].

## Figures and Tables

**Figure 1 ijms-23-14867-f001:**
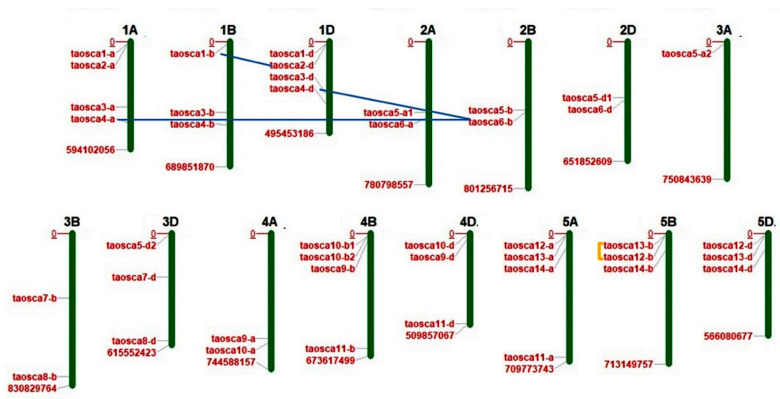
Chromosomal localization and duplication analysis of OSCA genes in Triticum aestivum. The TaOSCA genes are distributed on a total of 15 chromosomes of the A, B, and D subgenomes of T. aestivum. Blue lines point toward the existence of segmentally duplicated genes and yellow delineated shapes indicate the existence of tandemly duplicated gene pairs.

**Figure 2 ijms-23-14867-f002:**
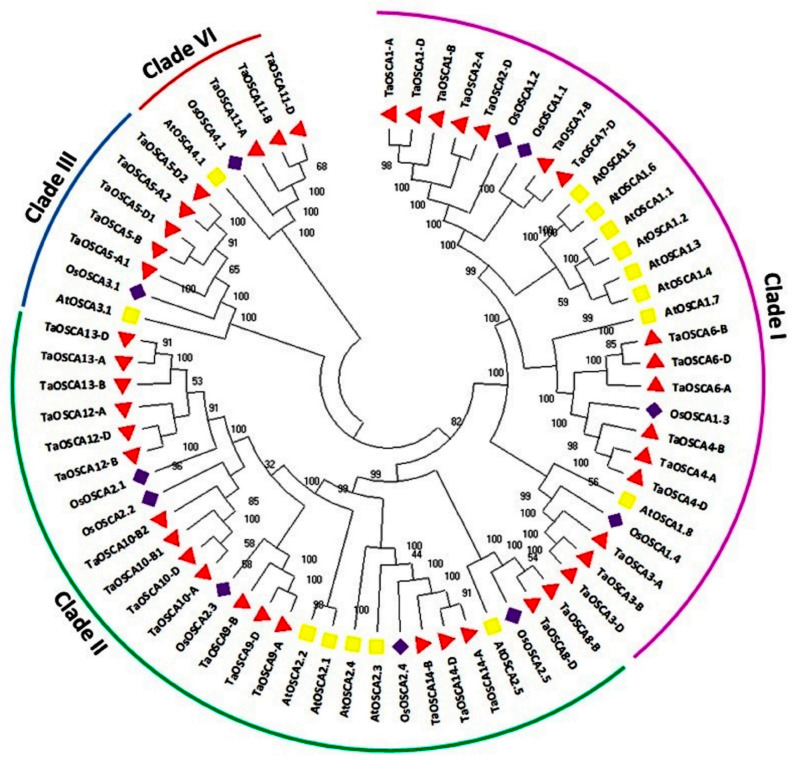
Phylogenetic tree analysis of A. thaliana, O. sativa, and T. aestivum using full-length peptide of OSCAs. The figure shows the phylogenetic tree of OSCA built by the neighbor-joining approach with 1000 bootstrap values using MEGA X software. The phylogenetic tree displays the clustering of the OSCA proteins of three plant species in different clades I–IV; each clade is colored differently.

**Figure 3 ijms-23-14867-f003:**
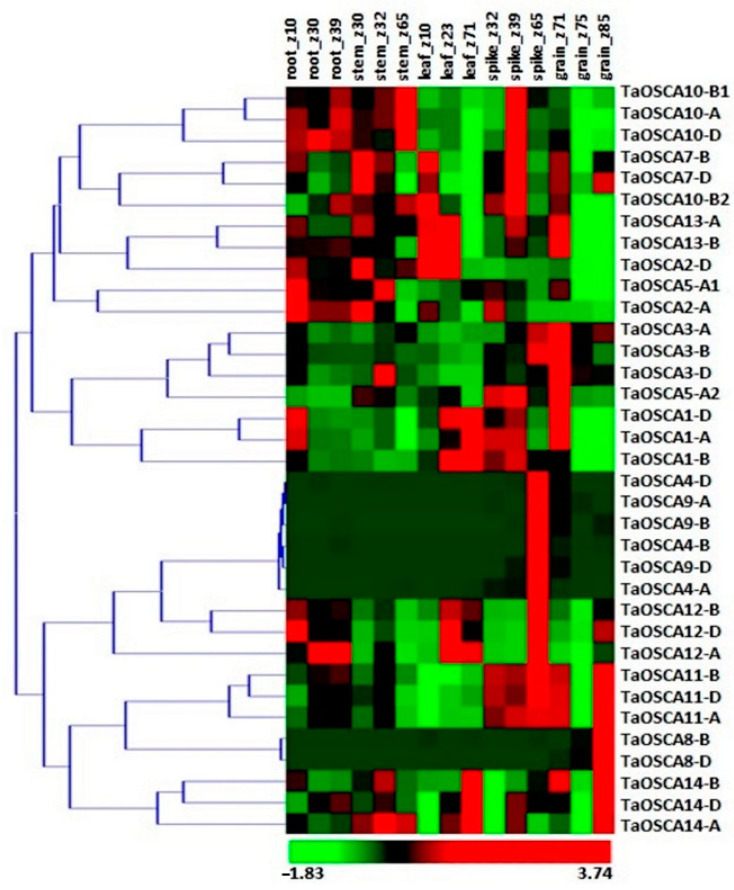
Expression profiling of TaOSCA genes of T. aestivum in various tissue and developmental stages. The heat map shows the clustering and expression profiling of OSCA genes of T. aestivum. The color bar depicts upregulated and downregulated expression with red and green color, respectively.

**Figure 4 ijms-23-14867-f004:**
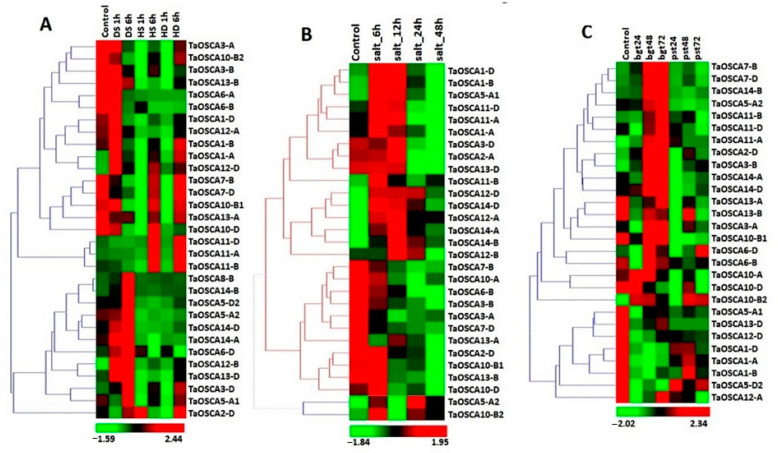
Expression profiling of OSCA genes of T. aestivum under abiotic and biotic stress conditions. Heat maps (**A**–**C**) show the expression profiles of two-fold up and down-regulated genes under (**A**) drought stress (DS), heat stress (HS), and HD (heat–drought), (**B**) salt stress, and (**C**) Blumeria graminis (Bgt) and Puccinia striiformis (Pst) infestation, respectively. The color bar shows high and low expressions with red and green color, respectively.

**Figure 5 ijms-23-14867-f005:**
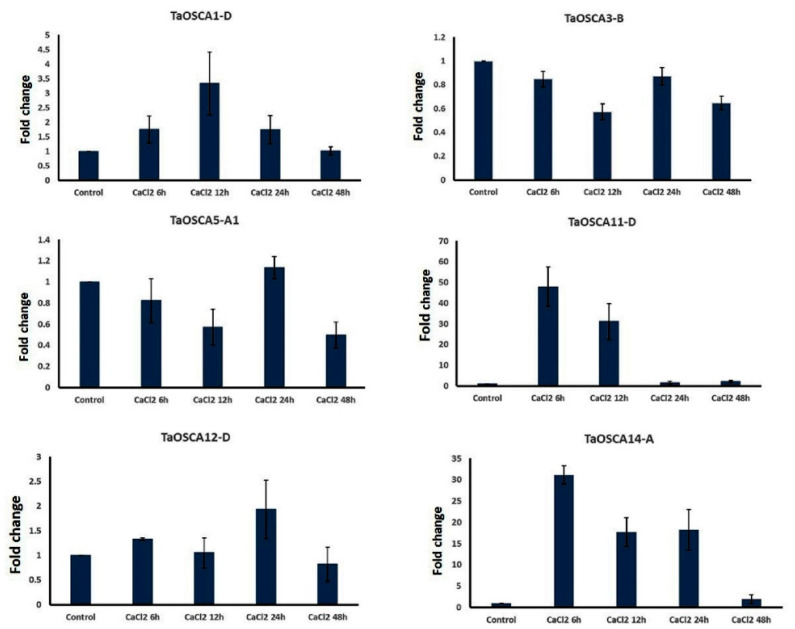
qRT-PCR analysis. Expression analysis of six TaOSCA genes was performed using qRT-PCR. Bar graphs show the expression trends of selected genes at 6, 12, 24, and 48 h of CaCl_2_ treatment. The bar graphs represent the fold change of each gene expression and the vertical lines indicate the standard deviation.

**Figure 6 ijms-23-14867-f006:**
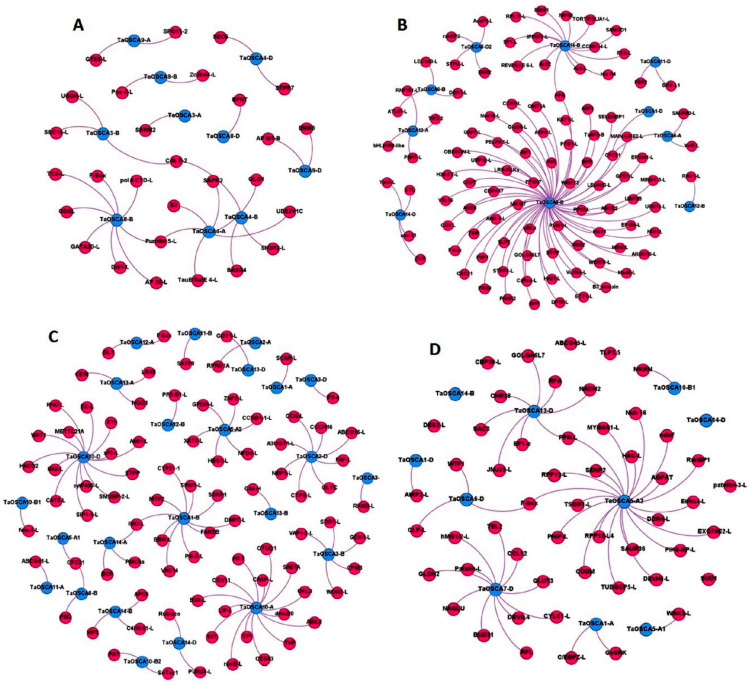
The interaction network analysis of OSCA genes with co-expressed genes was generated by using Gephi 0.9.1. The interaction network (**A**) shows co-expression in various tissues and developmental stages. The interaction network (**B**) shows the co-expression in the presence of HS, DS, and HS, (**C**) in salt, and (**D**) in biotic stress.

**Figure 7 ijms-23-14867-f007:**
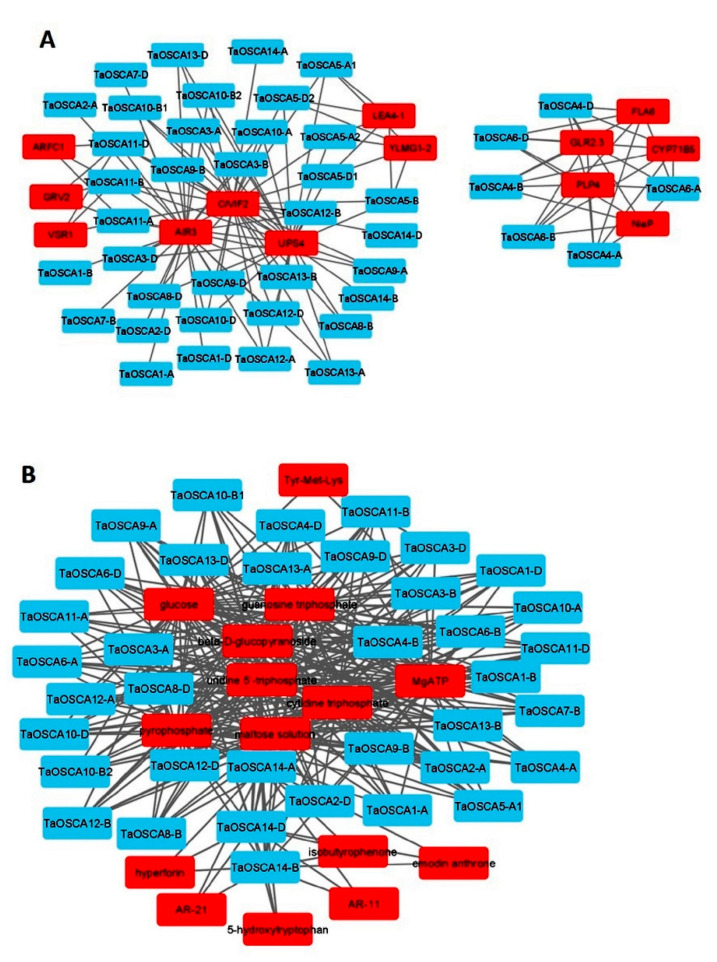
Interaction analysis of OSCA proteins using the STRING and STITCH servers. The networks (**A**) show the protein–protein interactions, where TaOSCA proteins are marked with blue and predicted interacting proteins are marked with red. The network (**B**) shows the protein–chemical interactions, where TaOSCA proteins are marked with blue and predicted interacting chemicals are marked with red. The networks were developed using the Cytoscape software.

**Figure 8 ijms-23-14867-f008:**
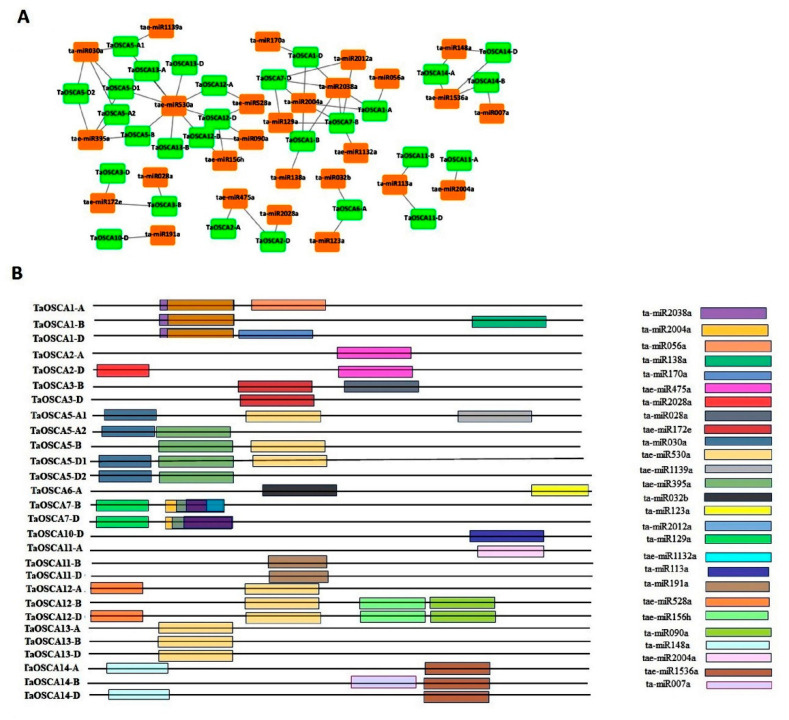
miRNA interaction analysis. The interaction network (**A**) of known miRNA of T. aestivum with TaOSCA genes. The prediction and generation of the network were performed by psRNAtarget tool and Cytoscape software, respectively. (**B**) shows the arrangement of miRNA target sites on TaOSCA genes. The colored boxes represent the predicted miRNAs of T. aestivum.

**Table 1 ijms-23-14867-t001:** The Ka/Ks ratios and divergence times of duplicated TaOSCA gene pairs.

Gene A	Gene B	Ka	Ks	Ka/Ks	T = Ks/2r	Selection Pressure
*TaOSCA4-A*	*TaOSCA6-B*	0.0885	0.5033	0.1758	13.5 MYA	Purifying
*TaOSCA1-B*	*TaOSCA2-D*	0.0697	0.2413	0.2886	22.2 MYA	Purifying
*TaOSCA12-B*	*TaOSCA13-B*	0.1297	0.3029	0.4282	32.9 MYA	Purifying

Ka, non-synonymous substitutions per non-synonymous site; Ks, synonymous substitutions per synonymous site; T, divergence time.

**Table 2 ijms-23-14867-t002:** Tajima’s relative rate tests of paralogous genes.

Group A	Group B	Outgroup	Nt	Na	Nb	χ^2^	*p*
*TaOSCA4-A*	*TaOSCA6-B*	*TaOSCA11-B*	93	10	6	1.00	0.31731
*TaOSCA1-B*	*TaOSCA2-D*	*TaOSCA5-B*	201	13	15	0.14	0.70546
*TaOSCA12-B*	*TaOSCA13-B*	*TaOSCA11-D*	112	13	11	0.17	0.68309

Nt, identical sites in all three sequences; Na, unique differences in sequence A; Nb, unique differences in sequence B.

## Data Availability

Not applicable.

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
