# Peer review of "OSCA Genes in Bread Wheat: Molecular Characterization, Expression Profiling, and Interaction Analyses Indicated Their Diverse Roles during Development and Stress Response"

_ijms, 2022, doi:10.3390/ijms232314867_

Round 1

Reviewer 1 Report

Kaur and coworkers analyzed the genome-wide expression and miRNA interactions of genes encoding pore-forming transmembrane proteins, namely hyperosmolality-gated calcium-permeable channels (OSCA) in bread wheat. They identified 42 OSCA genes and analyzed their phylogenetic origin, expression in developmental stages and stress conditions, co-expression partners and possible interactions with miRNA, other proteins and chemicals. The manuscript is too bulky with data based on predictions and meta-analyses. Because of this, the manuscript is very hard to read and looks less appealing.

The manuscript needs thorough revision by removing data which is not making much sense.

Specific comments:

1.       English is not smooth with unusual vocabulary. A thorough language editing could make the manuscript easy to read.

2.       Introduction can be improved with a more appealing rationale for the topic.

3.       Last paragraph of the introduction is too descriptive summarizing methods and results. This part can be reduced to a few key findings.

4.       Word ’in-silico’ should be used to make it convenient for readers.

5.       Line 101: ‘Physicochemical analyses..’ There are no physicochemical analyses done here. Use appropriate wordings.

6.       How was plasma membrane localization of all OSCA proteins determined? Is it based on transmembrane domains only?

7.       Line 165: “exhibited divergent expression patterns, suggesting the neo-functionalization”. It’s too blunt without wet lab validation. How could distinct expressions suggest neo-functionalization? There could be many possibilities. Duplicated genes may express to complement the primary gene or they may express in case the primary gene is not functioning.  

8.       What prompted the authors to analyze the expression of OSCA genes in abiotic and biotic stresses?

9.       Does co-expression

10.   Why miRNAs interaction analysis was carried out?

11.   As Tong et al., 2021, already reported the presence of 42 OSCA genes in the T. aestivum L. genome; why authors carried out this study? And what new information they fetch out. This should be included in the discussion.

12.   Discussion is redundant to results.  The key findings should be discussed instead of detailing the results.

13.   Figure quality needs improvements.

14.   Fig. 3 and 5, what do the range values in the color code represent?

15.   Fig. 6, Fold change with respect to what? It should be relative expression with respect to ARF1.

Author Response

Reviewer 1

Kaur and coworkers analyzed the genome-wide expression and miRNA interactions of genes encoding pore-forming transmembrane proteins, namely hyperosmolality-gated calcium-permeable channels (OSCA) in bread wheat. They identified 42 OSCA genes and analyzed their phylogenetic origin, expression in developmental stages and stress conditions, co-expression partners and possible interactions with miRNA, other proteins and chemicals. The manuscript is too bulky with data based on predictions and meta-analyses. Because of this, the manuscript is very hard to read and looks less appealing.

Response: We are grateful to the reviewer for the critical evaluation and valuable suggestion, which were very useful in the improvement of revised manuscript. We have reorganized the manuscript in such a way that it become easy to understand. 

The manuscript needs thorough revision by removing data which is not making much sense.

Response: We have removed comparative expression profiling of duplicated genes from the manuscript. And complete Ms has been revised properly as you can see in the correction highlighted Ms.

Specific comments:

  1. English is not smooth with unusual vocabulary. A thorough language editing could make the manuscript easy to read.

Response: Language has now been improved.

  1. Introduction can be improved with a more appealing rationale for the topic.

Response: Rewritten as suggested.

  1. Last paragraph of the introduction is too descriptive summarizing methods and results. This part can be reduced to a few key findings.

Response: Rewritten as suggested.

  1. Word ’in-silico’ should be used to make it convenient for readers.

Response: Suggested change have been done.

  1. Line 101: ‘Physicochemical analyses.’ There are no physicochemical analyses done here. Use appropriate wordings.

Response: The term “physicochemical” has been replaced with “in-silico protein characterization” in the manuscript

  1. How was plasma membrane localization of all OSCA proteins determined? Is it based on transmembrane domains only?

Response: The localization of all OSCA proteins was predicted using the Cello tool. In addition to it, the presence of transmembrane domain in each OSCA protein pointed toward their localization on plasma membrane.

  1. Line 165: “exhibited divergent expression patterns, suggesting the neo-functionalization”. It’s too blunt without wet lab validation. How could distinct expressions suggest neo-functionalization? There could be many possibilities. Duplicated genes may express to complement the primary gene or they may express in case the primary gene is not functioning.  

Response: This section has been removed from the revised manuscript as suggested above.

  1. What prompted the authors to analyze the expression of OSCA genes in abiotic and biotic stresses?

Response: Several earlier studies have shown the expression analysis of OSCA genes during stress conditions in various plant species such as Oryza sativa, Pyrus bretschneideri, Vigna radiata, Glycine max (Li et al., 2015, Gu et al., 2018, Zeng et al., 2020, Yin et al., 2021) etc. To study the effect of abiotic and biotic stresses on the OSCA genes of T. aestivum, we have performed their expression profiling under abiotic stresses (heat, drought and combined heat drought) and biotic stresses (two fungal pathogens including Blumeria graminis and Puccinia striiformis). 

  1. Why miRNAs interaction analysis was carried out?

Response: The miRNAs regulate the expression of several genes which involved in plant developmental processes and stress responses (Contreras-Cubas et al., 2012; Bai et al., 2018; Lin et al., 2015; Tang et al., 2016).  Recently, in an article, several miRNAs were reported which targets the OSCA genes of Hordeum vulgare and these miRNAs were responsive to growth and stresses, especially drought stress. With the purpose of prediction of targeted miRNAs of OSCA genes of T. aestivum and their involvement in plant biological processes and stress responses, we have performed miRNAs interaction analysis. Moreover, to study the involvement of TaOSCA proteins in plant metabolism and signalling processes, the interacted chemicals have been identified and analyzed. 

  1. As Tong et al., 2021, already reported the presence of 42 OSCA genes in the T. aestivumL. genome; why authors carried out this study? And what new information they fetch out. This should be included in the discussion.

Response: In our manuscript, we have discussed the TaOSCA genes in various directions to get a deep understanding of their evolution and functionality. To study the evolutionary history, we have performed duplication event analysis and Ka/Ks analysis and phylogenetic analysis to study the relationship with OSCA of dicots (Arabidopsis thaliana) and monocots (Oryza sativa and Triticum aestivum). For the study of functions, we have performed expression profiling in several tissue developmental stages and stress conditions. qRT-PCR was done to analyse the effect of CaCl2 stress on the OSCA genes. Moreover, we have done co-expression, miRNA, protein-protein and protein-chemical analysis have been performed in the current study. These analyses are not carried out by Tong et al., 2021.    

  1. Discussion is redundant to results.  The key findings should be discussed instead of detailing the results.

Response: Rewritten as suggested.

  1. Figure quality needs improvements.

Response: Improved in revised Ms.

  1. Fig. 3 and 5, what do the range values in the color code represent?

Response: It is normalized range from downregulation to upregulation. This normalized range automatically set up by the HCE tool.

  1. Fig. 6, Fold change with respect to what? It should be relative expression with respect to ARF1.

Response: Fig. 6 showed the Fold change of each gene with respect to control conditions.  The gene expression was normalized to 1 in controlled conditions.

Reviewer 2 Report

The authors are introducing a characterization of OSCA family gene in Triticum aestivum in this paper. For these, different data sets available are used. This is a predictive and descriptive paper which could be improved with some confirmative experiments. In addition, the paper is hard to follow, there is no clear line. I will recommend the authors to readjust the storyline of the paper going from general information to more detailed one.

Major comments:

Since the authors are using different data sets from different publications some information is needed to guide the reader in the better understanding of the statements. Probably, some type of standardization can be done. If not, at least, explain which tissue, developmental stage and conditions were used for each data set.

The authors use 6 random genes from the family to check their expression pattern under calcium treatment. This random selection is odd after presenting the expression pattern of 4 pair of genes (supposedly duplications). The authors are claiming that are describing possible functional diversity in these duplicated genes. I will suggest the authors to use these genes and do some confirmation experiments. I suggest the authors to do some qRT-PCR of these duplicated genes under Ca treatment and another abiotic stress in order to confirm their predictions.

All the information presented is very interesting but seems that there is no link between the sections. I will suggest the authors to introduce some sentences explaining why they are analyzing this.

I will suggest the author to change the title and adapt it better to the content of the paper.

Minor comments:

Line 101: What the authors mean by “physiochemical analyses”? Where are these analyses coming from?

Line 128: “insignificant” is not an appropriate statistical term; non significant.

Lines 158, 188, 239, 246: “etc.” If the authors does not want to mention all of them, they need to find the new expressions, for exemple: among them.

Section: comparative expression profiling of duplicated genes in tissue developed stages. Which is the data used for this section? The authors should clarify this.

Figure 4A and 4B: the profiles look identical, it is suspicious. Could the authors confirm these? Are TaOSCA4-A and TaOSCA4-D the same gene? If it is the reason, why the authors does not represent the 3 genes in the same graph?

Could the authors explain which is the porpoise of the miRNA data and the protein-chemical interaction sections?

Could the authors analyse if there is any coincidence between expression network showed in fig 7 and protein-protein interaction in fig 9?

Discussion:

line 367: “in consistent with” or inconsistent with?

line 381: “it was concluded” for whom?

line 411: “their crucial participation in both Ca homeostasis and signalling.” How the authors know that? Did they check the expression of the genes under any type of altered Ca signalling situation?

Author Response

The authors are introducing a characterization of OSCA family gene in Triticum aestivum in this paper. For these, different data sets available are used. This is a predictive and descriptive paper which could be improved with some confirmative experiments. In addition, the paper is hard to follow, there is no clear line. I will recommend the authors to readjust the storyline of the paper going from general information to more detailed one.

Response: We are grateful to the reviewer for the critical evaluation and valuable suggestion, which were very useful in the improvement of revised manuscript. We have reorganized the manuscript in such a way that it become easy to understand. 

Major comments:

Since the authors are using different data sets from different publications some information is needed to guide the reader in the better understanding of the statements. Probably, some type of standardization can be done. If not, at least, explain which tissue, developmental stage and conditions were used for each data set.

Response: To perform the expression profiling, the RNA seq data were produced from three development stages of five tissues, which are root_z10, root_z30, root_z39, stem_z30, stem_z32, stem_z65, leaf_z10, leaf_z23, leaf_z71, spike_z32, spike_z39, spike_z65, grain_z71, grain_z75, and grain_z85. These developmental stages are in accordance with Zadok scale proposed by the Dutch phytopathologist Jan C. Zadoks. This information is now included in the expression section of results in the revised manuscript with proper citation. It is also rephrased in Method section.

The authors use 6 random genes from the family to check their expression pattern under calcium treatment.

Response: By mistake, we have used the word “randomly” in the manuscript. It is replaced with correct description. We have selected 2 genes from different subclades of clade I and II, and one gene from each clade III and IV of phylogenetic tree for the qRT PCR analysis to cover all the divergent group of genes. 

This random selection is odd after presenting the expression pattern of 4 pair of genes (supposedly duplications). The authors are claiming that are describing possible functional diversity in these duplicated genes. I will suggest the authors to use these genes and do some confirmation experiments. I suggest the authors to do some qRT-PCR of these duplicated genes under Ca treatment and another abiotic stress in order to confirm their predictions.

Response: Reviewer has mixed the results of two section. Moreover, to better clarity and as suggested by the reviewer 1, the comparative expression section of duplicated genes has been removed from revised Ms. 

All the information presented is very interesting but seems that there is no link between the sections. I will suggest the authors to introduce some sentences explaining why they are analyzing this.

Response: Thank you for the positive responses. The revised Ms has been reorganized as suggested

I will suggest the author to change the title and adapt it better to the content of the paper.

Response: Title has been changed.

Minor comments:

Line 101: What the authors mean by “physiochemical analyses”? Where are these analyses coming from?

Response: Physico-chemical properties are the intrinsic physical and chemical characteristics of a substance. In our current study, we have analyzed physicochemical properties including length of amino acids, molecular weight and isoelectric point. We have replaced the term “physicochemical” with “in-silico protein characterization” in the manuscript as suggested by reviewer 1.

Line 128: “insignificant” is not an appropriate statistical term; non significant.

Response: Done

Lines 158, 188, 239, 246: “etc.” If the authors does not want to mention all of them, they need to find the new expressions, for example: among them.

Response: Revised as suggested

Section: comparative expression profiling of duplicated genes in tissue developed stages. Which is the data used for this section? The authors should clarify this. Figure 4A and 4B: the profiles look identical, it is suspicious. Could the authors confirm these? Are TaOSCA4-A and TaOSCA4-D the same gene? If it is the reason, why the authors do not represent the 3 genes in the same graph?

Response: This section has been removed as recommended by the reviewer 1. 

Could the authors explain which is the porpoise of the miRNA data and the protein-chemical interaction sections?

Response: The miRNAs regulate the expression of several genes which involved in plant developmental processes and stress responses (Contreras-Cubas et al., 2012; Bai et al., 2018; Lin et al., 2015; Tang et al., 2016).  Recently, in an article, several miRNAs were reported which targets the OSCA genes of Hordeum vulgare and these miRNAs were responsive to growth and stresses, especially drought stress. With the purpose of prediction of targeted miRNAs of OSCA genes of T. aestivum and their involvement in plant biological processes and stress responses, we have performed miRNAs interaction analysis. Moreover, to study the involvement of TaOSCA proteins in plant metabolism and signalling processes, the interacted chemicals have been identified and analyzed. 

Could the authors analyse if there is any coincidence between expression network showed in fig 7 and protein-protein interaction in fig 9?

Response: From both the co-expression and protein-proteins interaction analysis, we have predicted the interacting partners which were responsive to stresses, especially drought stress. Additionally, vegetable and reproductive development responsive interacting partners were found in both the analyses. The detail has been discussed in the discussion of revised Ms.

Discussion:

line 367: “in consistent with” or inconsistent with?

line 381: “it was concluded” for whom?

Response: corrected

line 411: “their crucial participation in both Ca homeostasis and signalling.” How the authors know that? Did they check the expression of the genes under any type of altered Ca signalling situation?

Response: The OSCA are pore forming calcium-permeable channels, which facilitates the transportation of calcium ions in response to adverse conditions. Several previous reports have shown the participation of OSCA channels in Ca homeostasis and signalling (Yaun et al., 2014; Hou et al., 2014; Thor et al., 2020). Further, the interaction of OSCA genes with various calcium binding proteins and other downstream signalling pathway and an induced expression of OSCA genes under CaCl2 stress indicated their roles in Ca homeostasis and signaling. The details has been now discussed in the revised Ms.

Round 2

Reviewer 2 Report

The authors did a great job taking into account the reviewers suggestions. The manuscript increased the quality considerably.  

Author Response

The authors did a great job taking into account the reviewers suggestions. The manuscript increased the quality considerably.  

Response: We are grateful for your positive recomendation.